# A Molecular Characterization of the Allelic Expression of the *BRCA1* Founder Δ9–12 Pathogenic Variant and Its Potential Clinical Relevance in Hereditary Cancer

**DOI:** 10.3390/ijms25126773

**Published:** 2024-06-20

**Authors:** Julieta Dominguez-Ortiz, Rosa M. Álvarez-Gómez, Rogelio Montiel-Manríquez, Alberto Cedro-Tanda, Nicolás Alcaraz, Clementina Castro-Hernández, Luis Bautista-Hinojosa, Laura Contreras-Espinosa, Leda Torres-Maldonado, Verónica Fragoso-Ontiveros, Yuliana Sánchez-Contreras, Rodrigo González-Barrios, Marcela Angélica De la Fuente-Hernández, María de la Luz Mejía-Aguayo, Ulises Juárez-Figueroa, Alejandra Padua-Bracho, Rodrigo Sosa-León, Gabriela Obregon-Serrano, Silvia Vidal-Millán, Paulina María Núñez-Martínez, Abraham Pedroza-Torres, Sergio Nicasio-Arzeta, Alfredo Rodríguez, Fernando Luna, Fernanda Cisneros-Soberanis, Sara Frías, Cristian Arriaga-Canon, Luis A. Herrera-Montalvo

**Affiliations:** 1Unidad de Investigación Biomédica en Cáncer, Instituto Nacional de Cancerología-Instituto de Investigaciones Biomédicas, Universidad Nacional Autónoma de México (UNAM), Avenida San Fernando No. 22 Col. Sección XVI, Tlalpan, Mexico City 14080, Mexico; juls.a.a@gmail.com (J.D.-O.); rogeliomontiel.bq@gmail.com (R.M.-M.); ccastroh7@yahoo.com.mx (C.C.-H.); luisebh1919@gmail.com (L.B.-H.); lauram.con.es@gmail.com (L.C.-E.); rodrigop@ciencias.unam.mx (R.G.-B.); ferlmgp@gmail.com (F.L.); 2Instituto Nacional de Cancerología, Universidad Nacional Autónoma de México (UNAM), Coyoacán, Mexico City 04510, Mexico; 3Clínica de Cáncer Hereditario, Instituto Nacional de Cancerología, Av. San Fernando No. 22 Col. Sección XVI, Tlalpan, Mexico City 14080, Mexico; rosamag2@hotmail.com (R.M.Á.-G.); ontiverosfvero@gmail.com (V.F.-O.); princeton_818_1@hotmail.com (Y.S.-C.); marcela_dfh@comunidad.unam.mx (M.A.D.l.F.-H.); maria_luz94@hotmail.com (M.d.l.L.M.-A.); alejandra.padua@gmail.com (A.P.-B.); osaleon.roy@gmail.com (R.S.-L.); gaby.obre96@gmail.com (G.O.-S.); vidals02@yahoo.com (S.V.-M.); pau_nunez@me.com (P.M.N.-M.); abraneet@gmail.com (A.P.-T.); 4Núcleo B de Innovación en Medicina de Precisión, Instituto Nacional de Medicina Genómica, Periférico Sur 4809, Arenal Tepepan, Tlalpan, Mexico City 14610, Mexico; acedro@inmegen.gob.mx; 5Novo Nordisk Foundation Center for Protein Research, Faculty of Health and Medical Sciences, University of Copenhagen, Blegdamsvej 3A, 2200 Copenhagen, Denmark; nicolas.alcaraz@cpr.ku.dk; 6Instituto Nacional de Pediatría, Insurgentes Sur No. 3700-C. Coyoacán, Mexico City 04530, Mexico; ledactorres@gmail.com (L.T.-M.); ehatlujf@gmail.com (U.J.-F.); alfrodriguezgomez@gmail.com (A.R.); sarafrias@biomedicas.unam.mx (S.F.); 7Natural Resource Ecology Laboratory, Colorado State University, Fort Collins, CO 80521, USA; sergio.nicasio@gmail.com; 8Departamento de Medicina Genómica y Toxicología Ambiental, Instituto de Investigaciones Biomédicas, Universidad Nacional Autónoma de México (UNAM), México City 04510, Mexico; 9Wellcome Trust Centre for Cell Biology, ICB, University of Edinburgh, Michael Swann Building, King’s Buildings, Max Born Crescent, Edinburgh EH9 3BF, UK; fda.cisneros@gmail.com; 10Tecnológico de Monterrey, Escuela de Medicina y Ciencias de la Salud, Monterrey 64710, Mexico

**Keywords:** hereditary breast and ovarian cancer, *BRCA1*, pathogenic variants, founder mutation, *BRCA1^Δ9–12^*, allele differential expression, isoform, nanopore sequencing

## Abstract

Hereditary breast and ovarian cancer (HBOC) syndrome is a genetic condition that increases the risk of breast cancer by 80% and that of ovarian cancer by 40%. The most common pathogenic variants (PVs) causing HBOC occur in the *BRCA1* gene, with more than 3850 reported mutations in the gene sequence. The prevalence of specific PVs in *BRCA1* has increased across populations due to the effect of founder mutations. Therefore, when a founder mutation is identified, it becomes key to improving cancer risk characterization and effective screening protocols. The only founder mutation described in the Mexican population is the deletion of exons 9 to 12 of *BRCA1* (*BRCA1^Δ9–12^*), and its description focuses on the gene sequence, but no transcription profiles have been generated for individuals who carry this gene. In this study, we describe the transcription profiles of cancer patients and healthy individuals who were heterozygous for PV *BRCA1^Δ9–12^* by analyzing the differential expression of both alleles compared with the homozygous *BRCA1* control group using RT–qPCR, and we describe the isoforms produced by the *BRCA1* wild-type and *BRCA1^Δ9–12^* alleles using nanopore long-sequencing. Using the Kruskal–Wallis test, our results showed a similar transcript expression of the wild-type allele between the healthy heterozygous group and the homozygous *BRCA1* control group. An association between the recurrence and increased expression of both alleles in HBOC patients was also observed. An analysis of the sequences indicated four wild-type isoforms with diagnostic potential for discerning individuals who carry the PV *BRCA1^Δ9–12^* and identifying which of them has developed cancer.

## 1. Introduction

The hereditary component of cancer contributes to 10% of all cancer cases globally. Among these, hereditary breast and ovarian cancer syndrome (HBOC) is the most prevalent, constituting 10% of all breast cancer (BrCa) cases and 20% of all ovarian cancer (OvCa) cases [1,2,3,4]. HBOC is linked to heterozygous and germline pathogenic variants (PVs) in cancer predisposition genes, with *BRCA1* and *BRCA2* being the most commonly affected genes [5,6]. Identifying individuals with PVs in HBOC has the following significant implications: (I) facilitating timely cancer diagnosis through high-risk screening methods (e.g., magnetic resonance imaging for breast cancer); (II) enabling targeted therapies, such as poly(ADP-ribose) polymerase (PARP) inhibitors, recognized as treatment options for major HBOC-related malignancies; (III) implementing cancer risk reduction strategies, including risk-reducing surgeries (mastectomy and/or salpingo-oophorectomy) or chemoprevention with agents such as tamoxifen or raloxifene for breast cancer; and (IV) identifying asymptomatic individuals at a high risk for cancer through cascade molecular diagnostics, thereby enabling the effective prevention of cancer morbidity and mortality [1,2,3,4,5,6,7]. This comprehensive approach underscores the importance of early detection, targeted interventions, and personalized risk management strategies in mitigating the impact of HBOC.

Hence, systematically studying PVs within HBOC genes is paramount, with a focus on understanding their implications for specific phenotypes. This entails exploring phenotype–genotype correlations in terms of tumor risks, age of onset, differential responses to pharmacological treatments, and drug resistance. Additionally, it is crucial to investigate how these variants differ among populations based on their ancestry. For instance, certain PVs may be prevalent within populations due to founder effects, which result in reduced genetic diversity and an increased frequency of specific genetic variants [8,9]. The identification of founder mutations facilitates the characterization of HBOC patients and enhances screening strategies to identify individuals and families harboring PVs [10,11].

Within the Mexican population, a single *BRCA1* founder PV has been documented: the exon 9–12 deletion (*BRCA1^Δ9–12^*) [1,11,12]. *BRCA1^Δ9–12^* arises from splicing between the AluSp element in intron 8 and the AluSx element in intron 12, resulting in a significant structural rearrangement and a loss of 15.4 kb of the BRCA1 gene [12]. This deletion induces a premature stop codon in the mRNA, suggesting its likely deleterious nature [12]. Moreover, the exon 9–10 deletion is the most prevalent PV identified in Mexico, associated with a heightened risk of high-grade epithelial ovarian cancer occurring at a younger age. The exon 9–10 deletion also exhibits an extended favorable response to the PARP inhibitor olaparib for recurrent ovarian cancer [1,11,12,13].

However, there is a dearth of studies elucidating the mechanisms underlying these clinical discrepancies compared to observations with other pathogenic variants (PVs). Specifically, there is a notable absence of research on the isoforms generated from both the *BRCA1* wild-type (WT) and *BRCA1^Δ9–12^* (Δ9–12) alleles. The evidence suggests that transcripts of BRCA1 PVs may evade nonsense-mediated mRNA decay (NMD) mechanisms via isoform splicing events or the upregulation of WT isoforms, thereby rescuing the homologous recombination (HR) response pathway involving BRCA1. This phenomenon has been associated with resistance to PARP inhibitors and platinum-based therapies [14,15,16,17,18,19]. However, it remains unknown whether the *BRCA1^Δ9–12^* mutation generates a translatable transcript or modulates the expression of WT isoforms.

This study aimed to characterize the transcript expression of both WT and Δ9–12 alleles in cancer patients (CaH) and healthy individuals (HH) heterozygous for the *BRCA1^Δ9–12^* PV, as well as their relation to the homozygous WT status of healthy individuals not carrying the PV (Ctrl). Our preliminary findings suggest that specific expression profiles may hold significance in the early detection of this founder PV in asymptomatic individuals, offering a cost-effective alternative via qPCR for diagnosing and monitoring asymptomatic individuals (Figure 1).

## 2. Results

### 2.1. Clinical Characteristics of the Patient Cohort

The characterization of the patient cohort was an important goal of this study and the starting point for establishing a correlation between transcription allele differences across the groups. The main clinical features were described in relation to tumor pathological characteristics, treatment, disease-free survival time, olaparib maintenance treatment, and recurrence (Table 1). The cohort comprised 11 female patients, all of whom were in the CaH group. One patient (CA11) lacked a description of her clinical characteristics, since her treatment was conducted at another institution and her clinical history was unavailable; her cancer diagnosis and *BRCA1* PV confirmation were corroborated. Among the following 10 patients, the age of primary tumor presentation ranged from 30 to 49 years. Four out of ten patients with available clinical histories were diagnosed with ovarian cancer as their first tumor, while the remaining six were diagnosed with breast cancer. The subsequent development of a secondary primary tumor was reported in three of the breast cancer patients and two of the ovarian cancer patients. The disease-free survival period ranged from 27 to 115 months, and six patients were reported to have cancer recurrence, none of which were breast–ovary related. Treatment for four of the six breast cancer patients included radiotherapy, and radioresistance was not reported in any of the patients (Table 1).

### 2.2. Analysis of BRCA1^WT^ and BRCA1^Δ9–12^ Transcript Expression in the Δ9–12 Heterozygous Groups and Controls

To describe and establish whether transcriptional differences were present between the allele expression among the patients who presented with HBOC and those who did not develop cancer and to compare it to that of the healthy Ctrl, we designed a specific set of primers to differentiate between the WT and Δ9–12 transcripts according to the size of the amplicons: 210 bp for the WT and 127 bp for the Δ9–12 transcripts (Figure 1A,B).

To ensure primer specificity, a primer mix reaction was utilized where the characteristic band pattern expected for each group could be visualized, one band in the Ctrl group showing the BRCA1 homozygous WT status, and two bands in both the CaH and HH groups exhibiting the *BRCA1* heterozygous WT and Δ9–12 alleles (Figure 1C,D). Sanger sequencing confirmed that exons 8–9 were spliced in the 210 bp WT amplicon and that exons 8–13 were spliced in the 127 bp Δ9–12 amplicon (Figure 1E,F). Transcript amplification using a single primer set was also performed to verify a single amplicon for each sample (Appendix A), along with the visualization of representative samples of each group in one electrophoresis gel. The samples were amplified by the primer mix reaction and the specific primers for the WT and Δ9–12 transcripts.

Once the specificity of our two sets of primers was verified, we used an RT–qPCR approach to evaluate the transcript expression. We found differences in the WT transcript expression between the groups (χ^2^ = 12.37; *p* = 0.002; d.f. = 2), except between the WT transcripts in the Ctrl (0.0094 ± 0.0028) and HH (0.0075 ± 0.0024) groups. The Δ9–12 allele also showed significant differences in gene expression (χ^2^ = 13.91; *p* = 0.003; d.f. = 3). The Δ9–12 allele showed a lower expression in HH (0.0043 ± 0.0017) and CaH (0.0026 ± 0.0013) than in their wild-type counterparts (HH: 0.0075 ± 0.0024; CaH: 0.0044 ± 0.0025). A significant difference was also observed between the Δ9–12 allele in CaH and the WT allele in HH (W = 2; *p* = 0.0003) (Figure 1G). The above patterns were not observed in any of the heterozygous groups, where the Δ9–12 allele expression remained below that of the WT allele; even though a slight increase was observed in the HH group, the difference was not significant. Equal WT transcript expression between HH and Ctrl individuals, as evaluated in lymphocytes, has not been reported, which makes this result an interesting antecedent for healthy PV heterozygous individuals.

### 2.3. Correlation of the WT and Δ9–12 CaH Transcript Expression Alleles with Clinical Characteristics

After identifying the differences in transcript expression for each allele in the CaH group, different clinical characteristics were evaluated to identify correlations suggesting new information about this PV and the patient’s clinical presentation. We compared previously related clinical characteristics to changes in allele-specific expression patterns for PVs in *BRCA1* [20]. Cancer recurrence was the only clinical characteristic showing significant differences for the WT and Δ9–12 alleles (Figure 2A,B). These findings provide a new observation not described for *BRCA1* PVs, adding to their genotype–phenotype description and concluding our evaluation of the transcription among HBOC patients.

### 2.4. Isoform Expression from the BRCA1^Δ9–12^ Allele

Another main objective of this work was to identify and study the isoforms and sequence heterogeneity in our group’s samples. To achieve this goal, we amplified the full-length *BRCA1* WT and Δ9–12 isoform sequences by endpoint PCR using a single primer set that was complementary to the adjacent region of the starting and stop codons (Figure 3A,B). Once the amplicons were obtained from each sample, our first approach to confirm the amplification of the *BRCA1* allele isoforms was to select representative samples from each group to be observed in a 1% gel (Figure 3C) and corroborated by Sanger sequencing of the purified bands, which corresponded to the expected weight of these transcripts and the joint of exons 8 and 9 for the WT strain and exons 8 and 13 for the Δ9–12 strain (Figure 3D).

We sequenced all the amplified isoforms from the samples using nanopore technology. The identified isoforms were analyzed using principal component analysis (PCA) of the top 79 expressed isoforms; the samples were separated by differentiation state, mostly along PC1. The expression patterns correlated and grouped the samples according to heterozygous Δ9–12 (CaH and HH) and homozygous WT (Ctrl) characteristics, as the PCA plot exhibited greater variance (35.49%) between these groups (Figure 4A). The CaH and HH groups clustered differently from each other on the PC3 axis, although with less variance (8.3%), as visualized in a 3D PCA plot (Figure 4B). A second analysis was undertaken using a heatmap of dissimilarity. Hierarchical clustering showed two different areas, where the Ctrl samples clustered within themselves with high similarity, and in the other panel, the CaH and HH samples correlated in another area (Figure 4C). This plot suggests that the transcriptional changes in isoform expression can distinguish between the Ctrl group and individuals with the *BRCA1^Δ9–12^* PV (CaH and HH), and confirms the differential expression of WT isoforms when the PV *BRCA1^Δ9–12^* is present.

### 2.5. Potential Predictors of PV and Cancer

Four isoforms were among the 79 *BRCA1* isoforms identified according to changes in expression. Two of them, isoform 1 (ENST00000352993.7) and isoform 2 (ENST00000484087.6), were overexpressed in samples with *BRCA1^Δ9–12^* (CaH and HH) and underexpressed in the Ctrl samples, allowing the differentiation between heterozygotes for PV *BRCA1^Δ9–12^* and healthy homozygotes for *BRCA1*. Isoforms 3 (ENST00000700082.1) and 4 (ENST00000618469.2) differentiated the samples within the CaH group from the HH group by being overexpressed in the HH samples and subexpressed in the CaH samples (Figure 4D). These results suggest that these isoforms could be used to differentiate the population of PV heterozygotes and identify those who develop cancer, making them potential biomarkers for identifying *BRCA1* PV patients.

## 3. Discussion

It is estimated that 10% of newly diagnosed breast and/or ovarian cancers are hereditary [21]. HBOC is an inherited predisposition to breast and/or ovarian cancers due to specific gene mutations [22]. The most important genes associated with hereditary breast and ovarian cancers are *BRCA1* and *BRCA2* [23,24,25]. Therefore, testing individuals for specific gene mutations and detecting PVs provides clinical information for genetic counseling, enabling directed oncological management and risk reduction surgeries [26,27]. These advantages expanded when PV detection increased for a founder mutation, such as *BRCA1^Δ9–12^*, in the population [9,10,12].

The benefits of identifying PVs in the population are not limited to the gene but also to the isoforms transcribed from it. The presence of specific variants, such as *BRCA1^2288delT^* and *BRCA1^185delAG^*, has been shown to have a substantial impact on patients who are heterozygous for these PVs, as resistance to PARP inhibitor treatment and platinum in the tumor via the rescue of the HR pathway has been explained through isoforms [17,18,28]. However, even though studying the mechanisms by which the isoforms provide molecular advantages in tumor resistance, and thus tumor survival, has been key to understanding their relevance in the context of cancer, their identification and description are the first fundamental steps in scaling this knowledge.

### 3.1. Group Identification through Specific Allele Transcript Expression

This study describes the changes in the expression of heterozygous allele transcripts of the pathogenic variant 9–12. We aimed to characterize the population that carries this PV and identify differences in its transcription, which may lead to significant clinical benefits from the observed correlations. The correlation between transcription and clinical associations in *BRCA1* PVs has been previously explored [20,29,30]. However, this work addresses this issue from an integrative perspective, including heterozygous healthy individuals, patients, healthy individuals, their non-PV counterparts, and allele-specific transcript assessment. The HH group and the individual allele evaluation provided information that allowed us to identify equal WT transcript expression between the two healthy groups, independent of their PV status, and observe the subexpression of the WT transcript only in cancer patients. Considering these results, the prospect of a biomarker for the follow-up of families with PV should be explored.

Given the limited cohort size and the absence of data that continue to inform us of changes in expression during cancer development, we cannot claim that gene compensation is the biological mechanism of this result. However, this approach should be explored as a possible strategy for healthy heterozygote cells to counter the haploinsufficiency of their WT allele [31,32] and avoid carcinogenesis, as has been proposed for biological models such as zebrafish, Drosophila, mice, human breast cancer cell cultures, and colorectal cancer tissues [32,33,34].

### 3.2. Isoform Pattern and Clinical Correlation

Among the isoforms that allow the study groups to be differentiated, isoforms 1, 2, and 4 have translational potential. Regardless, only isoform 4 was described as 100% similar to the *BRCA1* variant 1 protein. This suggests that isoform 4 can potentially rescue the HR repair pathway, as observed in other *BRCA1* VPs [15], possibly influencing the similarity observed in the allele-specific transcript analysis. Therefore, functional analyses need to be performed to investigate the effect of this isoform at the transcript and protein levels.

In summary, this work significantly contributes to describing the genotypic characteristics of the *BRCA1^Δ9–12^* PV at the transcriptional level and proposes two main strategies for diagnostic tools: the use of changes in transcript expression and the use of four isoforms with biomarker potential. To translate these results to clinical practice, the further evaluation of patients and their healthy relatives in a larger cohort and a prospective follow-up are needed to delineate when expression levels change between the two groups. Subsequent research into the biological mechanism by which WT allele expression changes when cancer occurs should be conducted. It is fundamental to continue describing the new isoforms identified during this project, not only for their translation potential but also to offer a breakthrough in the characterization of Hispanic populations.

## 4. Materials and Methods

### 4.1. Sample Collection

This study included a total of 40 RNA samples, 11 of which came from leukocytes of CaH of the Hereditary Cancer Clinic of the National Cancer Institute (INCan) in Mexico and nine from HH of the patients’ families; all 20 samples were heterozygous for PV Δ9–12. The *BRCA1^Δ9–12^* mutation was confirmed in all the participants, and complete gene sequencing confirmed no other mutation in either allele; these results were published by Fragoso in 2019. Familiar extensions for diagnosis and family consultation were followed at INCan. Informed consent was obtained from all groups. The *BRCA1* Ctrl yielded the remaining 20 RNA samples. All the samples were collected under the Ethics in Research Committee of the National Cancer Institute (CEI/1036/16). The sample selection workflow followed throughout the study is described in the Appendix A.

### 4.2. RNA Extraction

RNA was extracted from the cells using Direct-zol^TM^ RNA MiniPrep (ZYMO, Irvine, CA, USA, ref. R2052) according to the manufacturer’s instructions. Total RNA was quantified using a NanoDrop 3300 (Thermo Scientific, Waltham, MA, USA), and the OD260/280 ratio was determined. The RNA integrity number (RIN) was determined using an Agilent Bioanalyzer 2100 with an RNA 6000 Nano Assay (Agilent Technologies, Santa Clara, CA, USA).

### 4.3. cDNA Synthesis and Endpoint PCR Assay

Two different protocols were used, considering the purpose of the amplified transcripts. Protocol 1 aimed to obtain the amplicons of a partial region of the BRCA^WT^ and *BRCA1^Δ9–12^* transcripts for expression evaluation by RT–qPCR, and Protocol 2 aimed to amplify the complete sequence of the transcripts and their isoforms for analysis by nanopore sequencing.

#### 4.3.1. Protocol 1

cDNA synthesis was performed using a High-Capacity cDNA Reverse Transcription Kit (Thermo Scientific, Waltham, MA, USA, ref. 4368814) according to the manufacturer’s instructions.

The endpoint PCR was carried out with a total volume of 20 µL with 1X DreamTaq Buffer, 0.06 U of DreamTaq DNA Polymerase (Thermo Scientific, Waltham, MA, USA, ref. EP0701), a concentration of 0.4 µM dNTPs (Thermo Scientific, Waltham, MA, USA, ref. R0181), 0.4 µM of each primer (two sets of primers were used to amplify each transcript allele; detailed information is given in the primers section), and 1 μL of cDNA. The thermal cycling conditions were 94 °C for 5 min, followed by 40 cycles of 94 °C for 30 s, primer annealing at 59 °C for 30 s, primer extension at 72 °C for 45 s, and a final extension at 72 °C for 7 min. PCR products were resolved in a 1% agarose gel using electrophoreses for 45 min, and a 1 Kb ladder was used (Axygen, Central Ave Union City, CA, USA, ref AXY1016). This method corresponds to the results presented in Figure 1 and Figure 2. The band size comparison between alleles in the studied groups can be seen in Appendix A.

#### 4.3.2. Protocol 2

cDNA and PCR endpoints were synthesized from 500 ng of RNA using the SuperScript IV One-Step RT–PCR System (Thermo Scientific, Waltham, MA, USA, ref. 12594100) following the manufacturer’s instructions for protocol amplification and the BRCA1 Jong ex1/12 primer set, as detailed in the primers section. The conditions used for the thermal cycler program were as follows: reverse activation, 55 °C for 10 min; RT inactivation/initial denaturation, 98 °C for 2; amplification cycle, 36 cycles of 98 °C for 10 s, 60 °C for 30 s, and 72 °C for 3:30 min; and a final extension at 72 °C for 5 min. The final products were purified using a QIAquick PCR Purification Kit (QIAGEN, Germantown, MD, USA, ref. 28106) and resolved on a 1% agarose gel using electrophoresis for 45 min. A 1 Mb ladder was used (Rockland Immunochemicals, ref MB-204-0500). Representative amplicons from this protocol are presented in Figure 3 and Appendix A.

### 4.4. Primers

All primers were custom-designed by the researchers, except those used for amplifying the complete sequence of *BRCA1* (Table 2), which were obtained from the primer set published by Lucy C. de Jong et al. in 2017 [15].

### 4.5. RT–qPCR Assay

Ten microliters of RT–qPCR was performed using Maxima SYBR Green/ROX qPCR Master Mix (Thermo Scientific, Waltham, MA, USA, ref. K0222) and 0.16 µM of each primer. The thermal cycling conditions were 94 °C for 5 min, followed by 40 cycles of 94 °C for 30 s, primer annealing at 59 °C for 30 s, primer extension at 72 °C for 45 s, and a final extension at 72 °C for 7 min. Relative gene expression levels were assessed using ΔCT for the different sample groups, with normalization to the housekeeping gene *GAPDH*. This method is shown in the graphic in Figure 3.

### 4.6. GridION Library Preparation and Sequencing

We selectively included only PCR samples with concentrations exceeding 200 ng, comprising 10 control samples, seven carrier samples, and eight patient samples. The concentrations of these PCR products were determined using a Qubit dsDNA HS Assay Kit (Molecular Probes, Life Technologies, Eugene, OR, USA, ref. Q32851) according to the manufacturer’s guidelines. Each sample was subjected to amplification, yielding several amplicons of varying sizes that ranged from 200 to 6000 bp.

For sample preparation, we used the NEBNext Ultra II End Repair and dA-tailing Kit (New England BioLabs Inc., Ipswich, MA, USA). Native barcoding of the amplicons was conducted in a 20 µL reaction volume, which included 1.5 µL of DNA-repair amplicons, 2.5 µL of Native Barcode EXP-NBD104 and EXP-NBD114 (Oxford Nanopore Technologies, Oxford, Oxfordshire, UK), 10 µL of NEBNext Ultra II Ligation Master Mix (New England Biolabs, Ipswich, MA, USA), 0.5 µL of NEBNext Ligation Enhancer (New England Biolabs, Ipswich, MA, USA), and 5.5 µL of nuclease-free water. The mixture was incubated for 20 min at 20 °C, followed by 10 min at 65 °C. Subsequently, the barcoded amplicons were pooled.

For sequencing, we used the Ligation Sequencing Kit 1D (SQK-LSK110) according to the manufacturer’s protocol for adapter ligation. The library was then purified using AMPure XP beads and quantified using a Qubit dsDNA HS Assay Kit. We loaded 50 ng of the library onto an R.9.4.1 flow cell for sequencing on the GridION platform (Oxford Nanopore Technologies, Oxford, Oxfordshire, UK) over a period of 72 h. ONT MinKNOW software (Oxford Nanopore Technologies, Oxford, Oxfordshire, UK) was used to collect the raw sequencing data. We confirmed the availability of active pores upon receipt and immediately before the sequencing run. Detailed information is given in the primers section. This method is shown in Figure 4.

### 4.7. Bioinformatic Analysis

Fastq reads were trimmed for adapters and quality with Pychopper. The trimmed reads were then mapped to the genome GRCh38 built using minimap2 in splice mode using the parameters -ax splice --MD. The quantification of *BRCA1* transcripts (known and novel) was performed by mapping to the annotated gencode.v44 transcripts with salmon and the long-read parameter --ont --noLengthCorrection. Differential transcript expression was performed with DESeq2. An unsupervised method was applied.

### 4.8. Statistical Analysis

Considering the sample size, a normal distribution was assessed using the Lilliefors, Kolmogorov–Smirnov, and Shapiro–Wilk tests. The results confirmed that the samples did not show a normal distribution (*p* > 0.05). Therefore, we analyzed RT–qPCR differences among the three groups (cancer patients, healthy heterozygotes, and controls) using a nonparametric Kruskal–Wallis test. Furthermore, we employed a paired Wilcoxon test with a Bonferroni adjustment as a post hoc analysis to assess pairwise differences. The differential expression of both *BRCA1* transcripts in the clinical aspects of CaH was evaluated using the Wilcoxon test in conjunction with the Bonferroni adjustment. The differential statistic was established at *p* < 0.05, and the range of differences was indicated by the following values: * < 0.05, ** < 0.01, ***< 0.001.

## Data Availability

Sequencing data are available in Sequence Read Archive (SRA) with BioProject: PRJNA1113661.

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
