# Peer review of "A Molecular Characterization of the Allelic Expression of the BRCA1 Founder Δ9–12 Pathogenic Variant and Its Potential Clinical Relevance in Hereditary Cancer"

_ijms, 2024, doi:10.3390/ijms25126773_

Round 1

Reviewer 1 Report

Comments and Suggestions for Authors

This article focuses on molecular characterization of BRCA1 Founder Δ9-12 pathogenic variant allelic expression and its potential relevance and importance in Hereditary Cancer. This article highlights the importance of the founder mutation and how it can be studied to improve cancer risk characterization and effective screening protocol. The authors focused on the BRCA1Δ9-12 as it is the only founder mutation described in the Mexican population, and its transcriptional profiles has not been performed in individuals who carry it. The authors aimed to investigate the transcript expression of WT and Δ9-12 BRCA1 alleles in patients with Cancer and healthy individuals. The authors reported four isoforms that has a diagnostic potential for BRCA1 PV detection and cancer presentation. The study has revealed some interesting findings that have potential to be used as strategies for screening and early diagnosis of cancer or therapy resistance. However, I have some questions and concerns regarding some of the results. I would recommend this for publication after major review, after the authors have addressed the questions and concerns.

Supplementary Figures: The article mentions about supplementary figure s1 and s2, but I could not find it in any of the attachments. I’m sure this is a technical error and should be addressed.

Line 49: spelling error “funder”, as opposed to “founder”.

Line 90: Provide definition for the acronym “NMD”. It would be great for the readers to understand it.

Line 91: spelling error “homologous repay”. The authors have used HR as an acronym at multiple occasions, define the acronym.

Line 372: “RH” instead of “HR”.

Line 109: what do the author mean by “traductional”?

Line 173: It says figure 2g, but there is no figure 2g. It should be figure 1g instead.

Figure 1b: It says 168 bp instead of 127 bp.

Figure 3 (C and D): Is there a gel for the Healthy heterozygous sample as well? Did the author run gel for other samples (other than C3 and CA9)? If yes, it should be included in the supplementary figures at least. If these are not run on gel, I would suggest authors to run other samples and include that too. It would give more information about the transcript variants in these samples.

I have concern regarding the size of the PCR products in Figure 3(C and D). In figure 3C, the 5802 band runs near the topmost ladder band as opposed to Figure 3D, where it runs further down. The authors have not mentioned about these differences. Does it differ in gel %, different ladder, run time, amount of sample used?  I would suggest including about how this was done in materials and methods. Please include which ladders were used for these.

Author Response

For research article

Response to Reviewer #1 Comments

1. Summary

2. Questions for General Evaluation

Reviewer’s Evaluation

Response and Revisions

Does the introduction provide sufficient background and include all relevant references?

Can be improved

In accordance with the comments made by reviewer 1, the corresponding modifications were made.

Are all the cited references relevant to the research?

Yes

Is the research design appropriate?

Can be improved

Are the methods adequately described?

Must be improved

Are the results clearly presented?

Can be improved

Are the conclusions supported by the results?

Yes

3. Point-by-point response to Comments and Suggestions for Authors

This article focuses on molecular characterization of BRCA1 Founder Δ9-12 pathogenic variant allelic expression and its potential relevance and importance in Hereditary Cancer. This article highlights the importance of the founder mutation and how it can be studied to improve cancer risk characterization and effective screening protocol. The authors focused on the BRCA1Δ9-12 as it is the only founder mutation described in the Mexican population, and its transcriptional profiles has not been performed in individuals who carry it. The authors aimed to investigate the transcript expression of WT and Δ9-12 BRCA1 alleles in patients with Cancer and healthy individuals. The authors reported four isoforms that has a diagnostic potential for BRCA1 PV detection and cancer presentation. The study has revealed some interesting findings that have potential to be used as strategies for screening and early diagnosis of cancer or therapy resistance. However, I have some questions and concerns regarding some of the results. I would recommend this for publication after major review, after the authors have addressed the questions and concerns.

Comments 1: Supplementary Figures: The article mentions about supplementary figure s1 and s2, but I could not find it in any of the attachments. I’m sure this is a technical error and should be addressed.

Response 1: We agree with this comment. Therefore, we have revised the supplementary figures and corrected the error in the appearance of figures S1 and S2. Supplementary table 1 was included as table 1 within the manuscript to improve the traceability of the results and to avoid confusion with the supplementary figures, which were separated in a new file. Please see line 173.

Comments 2: Correction of lines: Line 49: spelling error “funder” as opposed to “founder”. Line 90: Provide definition for the acronym “NMD”. It would be great for the readers to understand it. Line 91: spelling error “homologous repay”. The authors have used HR as an acronym at multiple occasions, define the acronym. Line 372: “RH” instead of “HR”. Line 109: what do the author mean by “traductional”?. Line 173: It says figure 2g, but there is no figure 2g. It should be figure 1g instead. Figure 1b: It says 168 bp instead of 127 bp.

Response 2: Thank you for pointing this out. We have corrected the text, figure reference, and figure congruency errors. Please see lines 50 (correcting line 49), 116 (correcting line 90), and 108 (defining the HR acronym), “homologous repay” and “traductional” are no longer in the text, Figure 1g has been corrected in line 189 and the number 127 bp has been corrected in Figure 1b in the line 196.

Comments 3: Figure 3 (C and D): Is there a gel for the Healthy heterozygous sample as well? Did the author run gel for other samples (other than C3 and CA9)? If yes, it should be included in the supplementary figures at least. If these are not run on gel, I would suggest authors to run other samples and include that too. It would give more information about the transcript variants in these samples.

Response 3: We agree that Figure 3 (C and D) does not clearly expose the difference between the weights of the bands corresponding to the major isoforms of both alleles and that it would enrich to include a sample of healthy heterozygotes. We modified the figure with a gel in which the samples are viewed in a single run with the molecular weight marker better resolved. Please see lines 266 for the figure and 275-278 for the description.

We have also included in the supplementary figures a gel with the control samples used for sequencing, for which we still have material. However, due to the difficulty and scarcity of samples from patients and healthy heterozygotes, all material was prioritized for use in sequencing. Therefore, we have no more material to include in an electrophoresis gel.

Comments 4: I have concern regarding the size of the PCR products in Figure 3(C and D). In figure 3C, the 5802 band runs near the topmost ladder band as opposed to Figure 3D, where it runs further down. The authors have not mentioned about these differences. Does it differ in gel %, different ladder, run time, amount of sample used?  I would suggest including about how this was done in materials and methods. Please include which ladders were used for these.

Response 4: We hope the changes made to Figure 3 can resolve the doubts generated by the previous version. In the materials and methods section, we have included information on gel percentage, run time, and the size of the ladder used in the gels. Please see lines 262, 483-484 and 495-496.

4. Response to Comments on the Quality of English Language

Point 1: English language fine. No issues detected.

Response 1: Not Apply

5. Additional clarifications.

The project number for the nanopore sequencing data has been added to the line 542, which will be available once the article has been accepted.

Reviewer 2 Report

Comments and Suggestions for Authors

The manuscript presents a very good scientific problem, but the presentation of the context itself is catastrophically poor. Below, I have identified key areas that need improvement.

1.      The introduction of this manuscript extensively describes the prevalence and risk factors of hereditary breast and ovarian cancer syndrome (HBOC). However, while the manuscript presents important facts about the genetic nature of the syndrome and the most common mutations, the introduction section should be supplemented to better understand the purpose and context of the research being conducted. For example, it would be beneficial to provide a broader research objective and its significance for clinical practice or possible methodological approaches. The introduction should also more clearly emphasize how this study impacts HBOC patients and what specific clinical or practical benefits it may have.

2.      The bibliography consists of 32 sources, of which only 5 are from 2021 or more recent. I suggest adding at least 2-5 more articles from 2021 or newer to the list, for example:

Zhao, L., Lynch, L. & Eiriksson, L. Information needs of Lynch syndrome and BRCA 1/2 mutation carriers considering risk-reducing gynecological surgery: a qualitative study of the decision-making process. Hered Cancer Clin Pract 2024, 22, 5. https://doi.org/10.1186/s13053-024-00278-4

3.      Additionally, the introduction of the manuscript could be improved by including a brief summary of the key findings, which can be based on the discussed research results. This will help the reader connect the presented information with subsequent sections of the manuscript.

4.      The ultimate aim of the introduction should be to summarize the specific questions researchers aim to address and how this may impact the advancement of scientific knowledge or have practical implications in clinical practice. This will strengthen the context provided at the beginning of the manuscript and prepare the reader for further reading with a clear understanding of what information will be presented later in the text.

5.      The manuscript descends into complete chaos of information. Results are presented without any clearly articulated methodology. Meanwhile, the section intended to outline what is being investigated and the techniques used to support the study is presented last in the manuscript. Reorganize the structure of the manuscript fundamentally.

6.      No justification is provided for why a non-parametric method was chosen for statistical analysis, despite its lower accuracy compared to parametric methods. If this is related to assumptions about the distribution of data, then it should be explicitly stated, for example:

Arnastauskaite, et al. A New Goodness of Fit Test for Multivariate Normality and Comparative Simulation Study. Mathematics 2021, 9, 3003. https://doi.org/10.3390/math9233003

7.      The discussion section should be carefully structured to make it easier to understand the main themes and conclusions. Subheadings or bullet points could be used to reflect the key discussed results or aspects of the topic, making it clear to the reader what information will be presented.

8.      To better understand the results obtained in the study, it is important to discuss their clinical and practical significance. This includes a discussion on how the obtained results can be beneficial to clinical practice, patients, and their families, as well as how they can be applied in diagnostic and treatment strategies.

9.      Furthermore, it would be beneficial to provide a broader perspective on how the obtained results can contribute to the advancement of scientific knowledge within the research community and how they can be utilized in future research or practice. This will enhance the significance of the manuscript and justify why these research findings are important.

10.  Finally, it is recommended to summarize the main conclusions and highlight further research directions or potential avenues for future investigation, which could continue the research. This will help to conclude the manuscript with a clear message about achievements and potential future research paths.

Comments on the Quality of English Language

The language level is scientific and technical, which is expected and acceptable in a scientific context. However, sometimes the text could be slightly tighter and more effective. For example, some sentences could be shortened or restructured to make them easier to read.

Author Response

For research article

Response to Reviewer #2 Comments

1. Summary

2. Questions for General Evaluation

Reviewer’s Evaluation

Response and Revisions

Does the introduction provide sufficient background and include all relevant references?

Must be improved

In accordance with the comments made by reviewer 2, the corresponding modifications were made.

Are all the cited references relevant to the research?

Can be improved

Is the research design appropriate?

Must be improved

Are the methods adequately described?

Must be improved

Are the results clearly presented?

Can be improved

Are the conclusions supported by the results?

Must be improved

3. Point-by-point response to Comments and Suggestions for Authors

The manuscript presents a very good scientific problem, but the presentation of the context itself is catastrophically poor. Below, I have identified key areas that need improvement.

Comments 1: The introduction of this manuscript extensively describes the prevalence and risk factors of hereditary breast and ovarian cancer syndrome (HBOC). However, while the manuscript presents important facts about the genetic nature of the syndrome and the most common mutations, the introduction section should be supplemented to better understand the purpose and context of the research being conducted. For example, it would be beneficial to provide a broader research objective and its significance for clinical practice or possible methodological approaches. The introduction should also more clearly emphasize how this study impacts HBOC patients and what specific clinical or practical benefits it may have.

Response 1: We agree that the introduction should be supplemented with information conveying the research objective and stating the scope and clinical significance of the research. Therefore, the introduction was modified to consider the above. Please see lines 66-118.

Comments 2: The bibliography consists of 32 sources, of which only 5 are from 2021 or more recent. I suggest adding at least 2-5 more articles from 2021 or newer to the list, for example:

Zhao, L., Lynch, L. & Eiriksson, L. Information needs of Lynch syndrome and BRCA 1/2 mutation carriers considering risk-reducing gynecological surgery: a qualitative study of the decision-making process. Hered Cancer Clin Pract 2024, 22, 5. https://doi.org/10.1186/s13053-024-00278-4

Response 2: We agree with the recommendation to include more recent sources to enrich the literature review and ensure that the context of our research is up-to-date. We will add 2 to 5 references from 2021 onwards, including the study by Zhao et al., 2024 that you mentioned, to reflect the most recent developments in the field. Please see lines 70-80, 566-568, 585-587, and 608-610.

Comments 3: Additionally, the introduction of the manuscript could be improved by including a brief summary of the key findings, which can be based on the discussed research results. This will help the reader connect the presented information with subsequent sections of the manuscript.

Response 3: Thank you for pointing this out. We included a brief summary of the key findings at the end of the introduction to better link the discussed results with the subsequent sections of the manuscript. This will help prepare the reader for the experimental details and results that follow. Please see lines 112-118.

Comments 4: The ultimate aim of the introduction should be to summarize the specific questions researchers aim to address and how this may impact the advancement of scientific knowledge or have practical implications in clinical practice. This will strengthen the context provided at the beginning of the manuscript and prepare the reader for further reading with a clear understanding of what information will be presented later in the text.

Response 4: We agree with this comment. The introduction was restructured entirely to contextualize the clinical implications of our study. Please see lines 102-111.

Comments 5: The manuscript descends into complete chaos of information. Results are presented without any clearly articulated methodology. Meanwhile, the section intended to outline what is being investigated and the techniques used to support the study is presented last in the manuscript. Reorganize the structure of the manuscript fundamentally.

Response 5: We agree with this comment. Therefore, the methods' presentation was restructured to achieve an orderly and clear understanding of their implementation in this project. Please see lines 417-453

The results' connection with each section of the methods employed has also been specified. Please see lines 435, 446-447, 460-461and 486.

The presentation of the results was restructured to improve understanding. Supplementary Table 1 was included as Table 1 within the manuscript to improve the traceability of the results; please see lines 139 and 173-180. The manuscript was modified to improve its fluency; please see lines 156-158, 171-178, 233-235, 240-241, 256-265, 287-290, and 298-300. We also improved the description of Figure 4 and D; please see lines 311-316.

Comments 6: No justification is provided for why a non-parametric method was chosen for statistical analysis, despite its lower accuracy compared to parametric methods. If this is related to assumptions about the distribution of data, then it should be explicitly stated, for example:

Arnastauskaite, et al. A New Goodness of Fit Test for Multivariate Normality and Comparative Simulation Study. Mathematics 2021, 9, 3003. https://doi.org/10.3390/math9233003

Response 6: We agree that it is necessary to support the selection of the use of non-parametric tests in our statistical analysis by including in the material and method section the tests performed to identify whether our samples conformed to a normal distribution. Please see lines 495-499.

Comments 7: The discussion section should be carefully structured to make it easier to understand the main themes and conclusions. Subheadings or bullet points could be used to reflect the key discussed results or aspects of the topic, making it clear to the reader what information will be presented.

Response 7: We agree that the discussion needed to be restructured to convey the main themes and conclusion. Subheadings were added to clarify the presentation of the information. Please see lines 335-396.

Comments 8: To better understand the results obtained in the study, it is important to discuss their clinical and practical significance. This includes a discussion on how the obtained results can be beneficial to clinical practice, patients, and their families, as well as how they can be applied in diagnostic and treatment strategies.

Response 8: We agree to enhance in the discussion the clinical benefit that the patients and their families could receive from the results of this study. Please see lines 359-366.

Comments 9: Furthermore, it would be beneficial to provide a broader perspective on how the obtained results can contribute to the advancement of scientific knowledge within the research community and how they can be utilized in future research or practice. This will enhance the significance of the manuscript and justify why these research findings are important.

Response 9: We agree on the need to propose how our results can be used by the scientific community and in future research. We have added this aspect to the discussion. 371-375.

Comments 10: Finally, it is recommended to summarize the main conclusions and highlight further research directions or potential avenues for future investigation, which could continue the research. This will help to conclude the manuscript with a clear message about achievements and potential future research paths.

Response 10: We agree to summarize the main conclusions and emphasize the perspectives and where future research could be directed to strengthen this work. Please see lines 387-396.

We sincerely hope that we have responded to all the suggestions and that the new modifications made to the manuscript are in accordance with what was requested. Thank you for the time invested in reviewing our work to improve it.

4. Response to Comments on the Quality of English Language

Point 1: Minor editing of English language required. The language level is scientific and technical, which is expected and acceptable in a scientific context. However, sometimes the text could be slightly tighter and more effective. For example, some sentences could be shortened or restructured to make them easier to read.

Response 1: We agree with this comment, therefore we submit the manuscript in the English language editing to the American journal of experts with the number 4942-A90C-DC17-9E94-75D5.

5. Additional clarifications

The project number for the nanopore sequencing data has been added to the line 542, which will be available once the article has been accepted.

Round 2

Reviewer 1 Report

Comments and Suggestions for Authors

This article focuses on molecular characterization of BRCA1 Founder Δ9-12 pathogenic variant allelic expression and its potential relevance and importance in Hereditary Cancer. This article highlights the importance of the founder mutation and how it can be studied to improve cancer risk characterization and effective screening protocol. The authors focused on the BRCA1Δ9-12 as it is the only founder mutation described in the Mexican population, and its transcriptional profiles has not been performed in individuals who carry it. The authors aimed to investigate the transcript expression of WT and Δ9-12 BRCA1 alleles in patients with Cancer and healthy individuals. The authors reported four isoforms that has a diagnostic potential for BRCA1 PV detection and cancer presentation. The study has revealed some interesting findings that have potential to be used as strategies for screening and early diagnosis of cancer or therapy resistance. I would like to thank the authors for their responses to the questions and concerns. This article looks more complete now and would recommend this for publication after minor corrections.

There are still some writing errors that needs to be addressed. For e.g., Line 153: 2.2 Analysis of “BRA1WT”, as opposed to BRCA1WT. Line 187: “wil-type”. Line 329: “BRCA PV”.

Fig 3.C: Lane 1 label for the gel. “Lader”.

Figure S4: Can authors include the labels for the ladder and the respective 5802 bp on the gel. It would be easier to follow the bands with the labels for the band sizes.

Author Response

For research article

Response to Reviewer #1 Comments

1. Summary

2. Questions for General Evaluation

Reviewer’s Evaluation

Response and Revisions

Does the introduction provide sufficient background and include all relevant references?

Yes

In accordance with the comments made by reviewer 1, the corresponding modifications were made.

Are all the cited references relevant to the research?

Yes

Is the research design appropriate?

Yes

Are the methods adequately described?

Yes

Are the results clearly presented?

Yes

Are the conclusions supported by the results?

Yes

3. Point-by-point response to Comments and Suggestions for Authors

This article focuses on molecular characterization of BRCA1 Founder Δ9-12 pathogenic variant allelic expression and its potential relevance and importance in Hereditary Cancer. This article highlights the importance of the founder mutation and how it can be studied to improve cancer risk characterization and effective screening protocol. The authors focused on the BRCA1Δ9-12 as it is the only founder mutation described in the Mexican population, and its transcriptional profiles has not been performed in individuals who carry it. The authors aimed to investigate the transcript expression of WT and Δ9-12 BRCA1 alleles in patients with Cancer and healthy individuals. The authors reported four isoforms that has a diagnostic potential for BRCA1 PV detection and cancer presentation. The study has revealed some interesting findings that have potential to be used as strategies for screening and early diagnosis of cancer or therapy resistance. I would like to thank the authors for their responses to the questions and concerns. This article looks more complete now and would recommend this for publication after minor corrections.

Comments 1: There are still some writing errors that needs to be addressed. For e.g., Line 153: 2.2 Analysis of “BRA1WT”, as opposed to BRCA1WT. Line 187: “wil-type”. Line 329: “BRCA PV”.

Response 1: Thank you for pointing this out, we agree and have corrected the writing errors of the manuscript. Please see line 153, 187, and 329.

Comments 2: Fig 3.C: Lane 1 label for the gel. “Lader”.

Response 2: Thank you for pointing this out. We have corrected the text “Ladder” in Figure 3. C. Please see the line 267.

Comments 3: Figure S4: Can authors include the labels for the ladder and the respective 5802 bp on the gel. It would be easier to follow the bands with the labels for the band sizes.

Response 3: We agree to include the molecular weight in the bands of the ladder to improve Figure S4.

4. Response to Comments on the Quality of English Language

Point 1: English language fine. No issues detected.

Response 1: Not Apply

5. Additional clarifications

There are no further clarifications to be made.

Reviewer 2 Report

Comments and Suggestions for Authors

When readers go through a scientific article, they should not encounter questions whose answers are provided in later sections. The authors should guide them smoothly and consistently throughout the description of the study. Place the "Materials and Methods" section before the "Results" section. Ensure to maintain the coherence of the narrative while making these rearrangements.

Author Response

For research article

Response to Reviewer #2 Comments

1. Summary

2. Questions for General Evaluation

Reviewer’s Evaluation

Response and Revisions

Does the introduction provide sufficient background and include all relevant references?

Yes

In accordance with the comments made by reviewer 1, the corresponding modifications were made.

Are all the cited references relevant to the research?

Yes

Is the research design appropriate?

Yes

Are the methods adequately described?

Must be improved

Are the results clearly presented?

Yes

Are the conclusions supported by the results?

Yes

3. Point-by-point response to Comments and Suggestions for Authors

Comments 1: When readers go through a scientific article, they should not encounter questions whose answers are provided in later sections. The authors should guide them smoothly and consistently throughout the description of the study. Place the "Materials and Methods" section before the "Results" section. Ensure to maintain the coherence of the narrative while making these rearrangements.

Response 1: We understand the importance of the proposed order in the presentation of the manuscript, considering changing the methodology before the results, however we follow the IJMS Microsoft Word template provided by the journal Manuscript Submission portal, in order to comply with the outline of their publications and so we find it necessary to maintain this scheme for the above reasons.

4. Response to Comments on the Quality of English Language

Point 1: English language fine. No issues detected.

Response 1: Not apply.

5. Additional clarifications

There are no further clarifications to be made.

Round 3

Reviewer 2 Report

Comments and Suggestions for Authors

The manuscript can be accepted.